# Compromised heat loss leads to a delayed ice slurry induced reduction in heat storage

**Thomas J. O'Brien**◯*, **Victoria L. Goosey-Tolfrey, Christof A. Leicht**◯*

Peter Harrison Centre for Disability Sport; School of Sport, Exercise and Health Sciences, Loughborough University, Loughborough, United Kingdom

* t.obrien@lboro.ac.uk (TJO); c.a.leicht@lboro.ac.uk (CAL)

**Data Availability Statement:** All relevant data are within the manuscript and its Supporting Information files.

## Abstract

Compromised heat loss due to limited convection and evaporation can increase thermal strain. We aimed to determine the effectiveness of ice slurry ingestion to reduce thermal strain following hyperthermia in a state of compromised heat loss. Twelve healthy males (age: 25 ± 4y) underwent hot water immersion to elevate rectal temperature ($T_{rec}$) by 1.82 ± 0.08°C on four occasions. In the subsequent 60-min of seated recovery, participants ingested either 6.8 g·kg$^{-1}$ of ice slurry (-0.6°C) or control drink (37°C) in ambient conditions (21 ± 1°C, 39 ± 10% relative humidity), wearing either t-shirt and shorts (2 trials: ICE and CON) or a whole-body sweat suit (2 trials: ICE-SS and CON-SS). $T_{rec}$ and mean skin temperature ($T_{sk}$) were recorded and a two-compartment thermometry model of heat storage was calculated. Heat storage was lower in ICE compared with CON at 20-40min (p ≤ 0.044, d ≥ 0.88) and for ICE-SS compared with CON-SS at 40–60 min (p ≤ 0.012, d ≥ 0.93). $T_{rec}$ was lower in ICE compared with CON from 30-60min (p ≤ 0.034, d ≥ 0.65), with a trend for a reduced $T_{rec}$ in ICE-SS compared with CON-SS at 40min (p = 0.079, d = 0.60). A greater $T_{sk}$ was found in ICE-SS and CON-SS compared with ICE and CON (p < 0.001, d ≥ 3.37). A trend for a lower $T_{sk}$ for ICE compared with CON was found at 20-40min (p ≤ 0.099, d ≥ 0.53), no differences were found for ICE-SS vs CON-SS (p ≥ 0.554, d ≤ 0.43). Ice slurry ingestion can effectively reduce heat storage when heat loss through convection and evaporation is compromised, relevant to those wearing personal protective equipment or those with compromised sweat loss. Compromised heat loss delays the reduction in heat storage, possibly related to ice slurry ingestion not lowering $T_{sk}$.

## Introduction

Elevations in core temperature ($T_{core}$) and heat storage can result from exposure to hot and/or humid environments and can be further increased by metabolic heat production [1]. This thermal strain can be amplified if heat loss is reduced due to clothing, such as personal protective equipment (PPE), inhibiting convection and/or sweat evaporation [2, 3]. Similarly, disruptions to neural pathways, as found in persons with a spinal cord injury [4], impair central sudomotor and vasomotor control and sweating below the level of lesion [5], affecting heat loss [6]. Elevated $T_{core}$ can impact exercise performance and recovery, including deterioration of physical capacity [7], deterioration in cognitive function [8, 9] or heat related illness [10]. These

**Funding:** This research was funded by the Peter Harrison Centre for Disability Sport (PHC), a research centre within the School of Sport, Exercise and Health Sciences at Loughborough University. The funders had no role in the study design, data collection and analysis, decision to publish, or preparation of the manuscript. The lead other, Thomas J O'Brien was receiving a salary from the PHC at the time of submission. No external financial support has been received for the preparation of this work.

**Competing interests:** The authors have declared that no competing interests exist.

negative aspects resulting from hyperthermia can be mitigated by cooling strategies, such as cold water immersion or ice vests, which can effectively lower $T_{core}$ [11]. However, the practicalities of such strategies (e.g., time availability, equipment costs, safety in certain populations, and accessibility) limit their use in applied settings.

In the context of sport and exercise performance, increasing body temperature, for example achieved via a warm-up, can enhance performance [12]. However, exceeding an optimal body temperature can limit performance [7]. Internal cooling by ice slurry ingestion has become a popular method to attenuate increases in $T_{core}$ prior to (pre-cooling) and during (per-cooling) exposure to hot and humid environments and/or exercise [13–15]. However, the effectiveness of ice slurry ingestion as a strategy to reduce $T_{core}$ in a hyperthermic state is under debate. On the one hand, half-time cooling [16] and cooling when in a hyperthermic state ($T_{core} > 39.6°C$) [10] can decrease $T_{core}$ and improve repeated sprint performance in the heat. Conversely, Iwata et al. reported that ice slurry ingestion does not alter $T_{core}$ when administered at a (lower) $T_{core}$ of 38.5°C [17]. Differences in study design, such as ice slurry dosage and timing, should be noted. For example, Naito et al. employed an intermittent sprint protocol where ice slurry was ingested throughout [16], whereas Iwata et al. gave one bolus at the end of a continuous exercise bout aimed at inducing hyperthermia [17]. However, they further provide a suggestion for a mechanistic explanation, arguing that the lack of a cooling effect may lie in ice slurry ingestion hindering the sweat response [17], thus blunting evaporative cooling potential [18, 19], and therefore not significantly affecting heat balance. This may hence mean that ice slurry ingestion could be most effective when sweating does *not* contribute to cooling, as in such conditions it would not offset the (non-existing) cooling effect of sweat evaporation. Ice slurry ingestion might therefore be of particular interest in situations in which the effect of evaporative cooling is limited.

Few studies have investigated the efficacy of ice slurry ingestion to reduce thermal strain when heat loss mechanisms are compromised. Per-cooling using ice slurry reduced gastrointestinal temperature at lower, but not moderate, exercise intensities when wearing a raincoat restricting heat loss in normothermic conditions [20], whilst ice slurry ingestion reduced $T_{core}$ in participants wearing firefighting PPE during a 30-minute walk in hot conditions [21]. However, ingesting slurries during an activity may not always be feasible or practical in both sporting or professional contexts, and the efficacy of ice slurries to reduce thermal strain in the recovery of hyperthermia when heat loss is compromised remains to be explored.

Therefore, the aim of the present study was to assess the cooling effectiveness of ice slurry ingestion in a state of compromised heat loss, induced by wearing a whole-body sweat suit *following* passive heat exposure. We assumed that ice slurry ingestion would significantly reduce thermal strain in a state of compromised heat loss when compared with a state of normal heat loss.

## Methods

### Ethical approval

All experimental procedures were granted by the Loughborough University ethics committee, which were in line with the Declaration of Helsinki (review reference: 2020-1208-2037). Participants provided written informed consent and completed a health screening questionnaire prior to participating in any experimental procedures.

### Participants

Between 16th March 2021 - 2$^{nd}$ June 2021, twelve male able-bodied participants volunteered to take part in the study (age: 25 ± 4 y, body mass: 84.5 ± 6.6 kg, stature: 181.6 ± 7.2 cm, body

mass index: $25.7 \pm 2.5$ kg·m$^{-2}$, sum of 8 skinfolds: $91.4 \pm 28.4$ mm). Participants were asked to refrain from alcohol consumption or partaking in any strenuous activity 24 h prior to each visit and were asked to maintain a consistent routine during the day before each visit. A 24-h food diary was completed prior to the first visit, and the diet was replicated for subsequent visits.

## Experimental design

Four experimental trials were separated by at least 48 h in a randomized, counterbalanced order. During each trial, participants were passively immersed in hot water maintained at ~39.5˚C controlled by four thermistors (Grant Instruments, Cambridge, UK) placed at the top and bottom of the immersion bath. Before participants entered the hot water bath, they were instrumented, and baseline measures were collected after 30 min of seated rest. Participants were immersed in the bath until the target $T_{core}$ was reached (1.85˚C change from baseline ($n = 10$) or an absolute value of 39.35˚C if this came first (ethical limit, $n = 2$)), total immersion duration of was $58.2 \pm 11.2$ min. This was followed by a 60-min recovery phase consisting of passive seated rest in ambient conditions ($20.9 \pm 1.2$˚C, $39 \pm 10\%$ relative humidity, RH) which did not differ between trials ($p = 0.966$, $p = 0.795$ for temperature and RH respectively). Upon exiting the bath, participants rested seated and received two unflavoured aliquots (totalling 6.8 g·kg$^{-1}$) of either ice slurry drink ($-0.6 \pm 0.1$˚C; ICE) or a thermoneutral control drink (water at $37.0 \pm 0.3$˚C; CON) to ingest within 15 minutes at 5 and 20 min of the recovery phase. To produce the ice slurry drink, small cubes of ice were blended with water (4˚C). This post bath rest period was performed wearing shorts and T-shirt, creating a condition of uncompromised heat loss, or a sweat suit (SS), creating a condition of compromised heat loss. The SS was made with 98% polyester and 2% nylon and was available in three sizes, to fit the participants and only left feet, hands and the head exposed. The combination of drink type and clothing worn resulted in four conditions: ice slurry, no SS (ICE), control drink, no SS (CON), ice slurry, SS (ICE-SS), control drink, SS (CON-SS).

## Physiological assessments

Participants provided a urine sample upon arrival to the laboratory, which was analysed for osmolality using a hand-held portable analyser (Osmocheck, ViTech Scientific, UK) to confirm adequate hydration prior to commencing each trial. After that, for one of four of the trials, participants' skin fold thickness (mm) was taken from 8 sites (bicep, tricep, subscapularis, iliac crest, supraspinale, abdominal, thigh, calf) by a certified International Society for the Advancement of Kinanthropometry (ISAK) practitioner.

Rectal temperature ($T_{rec}$) was recorded with a probe that was self-inserted 10 cm beyond the anal sphincter (YSI 400 series; YSI, Yellow Springs, USA), skin temperature ($T_{sk}$) was recorded using iButtons (DS1922T, Maxim Integrated Products Inc., USA) placed on the forehead, bicep, chest, front thigh and medial calf to estimate mean $T_{sk}$ respectively [22]. Heat storage was calculated using the following formula [23]:

$$\text{Heat Storage (J} \cdot \text{g}^{-1}) = (0.8 \cdot \Delta T_{rec} + 0.2 \cdot \Delta T_{sk}) \cdot c_b$$

where $c_b$ is the specific heat capacity of the body tissue (3.49 J·g$^{-1}$°C$^{-1}$) and $\Delta T_{rec}$ and $\Delta T_{sk}$ represent changes in $T_{rec}$ and mean $T_{sk}$ from immediately prior to the immersion period.

Fully instrumented nude body mass was recorded 5 min prior to the passive heating phase, upon towel drying following passive heat exposure and following the 60-min recovery phase. Changes in body mass in response to hot water immersion and to recovery were used to estimate whole body sweat loss, correcting for fluid consumption and urine output.

### Thermal sensation

Thermal sensation was subjectively rated every 10 min throughout the recovery phase. The thermal sensation scale comprised of categories ranging from 0 ("unbearably cold") to 8 ("unbearably hot") in 0.5 increments [24].

### Statistical analysis

All data were analysed using the Statistical Package for Social Sciences (version 27; SPSS Chicago, IL). Data are expressed as mean ± SD unless otherwise stated. Normality and sphericity were checked using the Shapiro-Wilk test and Mauchly's test, respectively. A one-way repeated measures analysis of variance (ANOVA) was used to assess urine osmolality and baseline thermoregulatory measures between conditions. Analysis of $T_{rec}$, mean $T_{sk}$, heat storage, and thermal sensation during the recovery phase was performed using a 2 x 2 x 7 (SS condition x drink x time) repeated measures ANOVA. When a significant interaction effect was observed, and for data analysis between trials at one time point, one-way repeated measures ANOVAs were performed, P values of this post-hoc analyses were not adjusted for multiple comparisons [25]. Where assumptions of sphericity were violated, a Greenhouse-Geisser correction was applied. Significance was accepted at $p \leq 0.05$ with trends defined as $p = 0.05$–0.10. When referring to multiple outcomes, the less-than operator ($<$) is used. For pairwise comparisons between conditions, effect sizes were expressed using Cohen $d$, with the magnitude of effect size classed as small (0.20), moderate (0.50) and large (0.80) [26].

## Results

### Resting measurements

Urine osmolality and resting body mass, $T_{rec}$, mean $T_{sk}$, room temperature and relative humidity did not differ between trials (Table 1). Due to personal discomfort, three participants were unable to complete their prescribed ingestion volume of ice slurry drink during ICE ($n = 2$, ingested volume 95% of target) and ICE-SS ($n = 3$, ingested volume 92% of target), however, drink volume was kept identical in subsequent trials and thus drink volume did not differ between trials. All participants arrived in a hydrated state (urine osmolality $<700$ mOsm [27]).

### Recovery phase

**Thermoregulatory measures, heat storage and thermal sensation.**   At the start of the recovery phase, $T_{rec}$, mean $T_{sk}$ and thermal sensation did not differ between conditions ($p = 0.935$, 0.567 and 0.153, respectively).

**Table 1. Resting measures and ambient conditions.**

| | ICE | CON | ICE-SS | CON-SS | *P value* |
|---|---|---|---|---|---|
| Urine Osmolality (mOsm·kg⁻¹) | 467.5 ± 120.8 | 485.8 ± 174.4 | 498.3 ± 150.1 | 495.8 ± 153.1 | *0.501* |
| Body mass (kg) | 84.2 ± 6.6 | 84.5 ± 6.6 | 84.2 ± 6.3 | 84.1 ± 6.5 | *0.168* |
| $T_{rec}$ (˚C) | 37.4 ± 0.3 | 37.4 ± 0.3 | 37.4 ± 0.2 | 37.4 ± 0.2 | *0.982* |
| Mean $T_{sk}$ (˚C) | 30.4 ± 1.1 | 30.4 ± 1.3 | 30.6 ± 1.3 | 30.6 ± 1.7 | *0.858* |
| Room Temperature (˚C) | 20.8 ± 1.3 | 20.9 ± 1.4 | 21 ± 1.4 | 21.0 ± 1.1 | *0.819* |
| Relative Humidity (%) | 41 ± 10 | 40 ± 11 | 38 ± 9 | 38 ± 9 | *0.593* |

ICE, ice slurry, no sweat suit; CON, control drink, no sweat suit; ICE-SS; ice slurry, sweat suit; CON-SS, control drink, sweat suit. $T_{rec}$, rectal temperature; $T_{sk}$, skin temperature.

During the 60-min recovery, there was a significant SS-condition x time and drink x time interaction for heat storage ($p < 0.001$). Post-hoc analysis revealed heat storage was lower from 10 min onwards in ICE and CON compared with ICE-SS and CON-SS ($p \leq 0.047$, $d \geq 0.80$; Fig 1A). However, the ice-slurry induced reduction in heat storage was delayed in the SS trials: whilst heat storage was lower for ICE compared with CON at 20 to 40 min ($p \leq 0.044$, $d \geq 0.88$), heat storage was lower for ICE-SS compared with CON-SS at 40 to 60 min ($p \leq 0.012$, $d \geq 0.93$).

There was a significant main effect of time ($p < 0.001$) and a drink x time interaction ($p = 0.020$) for $T_{rec}$. Post-hoc analysis revealed that $T_{rec}$ was lower in ICE compared with CON from 30 to 60 min ($p \leq 0.034$, $d \geq 0.53$; Fig 1B), with the reduction in $T_{rec}$ in ICE-SS compared with CON-SS approaching significance with a moderate effect size at 40 min ($p = 0.079$, $d = 0.61$). Further moderate and large effects sizes showed ice slurry induced reductions for both ICE vs CON ($d = 0.53$–0.80) and ICE-SS vs CON-SS ($d = 0.53$–0.61) (Fig 1).

There was a significant SS-condition x time interaction for both mean $T_{sk}$ ($p < 0.001$, Fig 1C), mean $T_{sk}$ was lower in ICE and CON compared with ICE-SS and CON-SS from 10 min onwards ($p < 0.001$). A reduction in mean $T_{sk}$ for ICE compared with CON approached significance at 20 to 40 min with moderate effect sizes ($p \leq 0.099$, $d \geq 0.53$); mean $T_{sk}$ did not differ between ICE-SS and CON-SS throughout the recovery phase ($p \geq 0.554$, $d \leq 0.43$).

There was a main effect for time, and interaction effects (SS-condition x time and drink x time) for thermal sensation ($p = 0.001$, $p = 0.032$ and $p = 0.018$; Fig 1D). Thermal sensation was greater in CON-SS than CON ($p = 0.006$, $d = 0.57$), but did not differ between ICE-SS and ICE ($p = 0.130$, $d = 0.24$). Whole body sweat loss during immersion and recovery did not differ between conditions (Table 2).

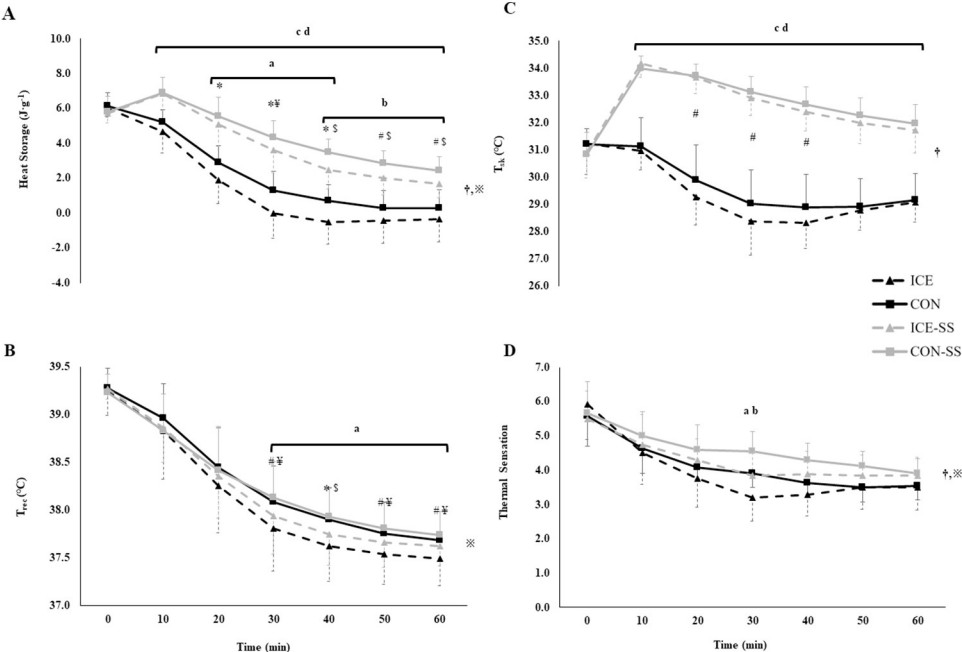

**Fig 1.** Heat storage (A), rectal temperature ($T_{rec}$; B), skin temperature ($T_{sk}$: C) and thermal sensation (D) in the recovery phase from hyperthermia. ICE (ice slurry, no sweat suit), CON (control drink, no sweat suit), ICE-SS (ice slurry, with sweat suit), CON-SS (control drink, with sweat suit). Significant interactions: [†]SS-condition x time, [※]drink x time; significant differences at individual time points: [a]ICE vs CON, [b]ICE-SS vs CON-SS, [c]CON vs CON-SS, [d]ICE vs ICE-SS (p < 0.05), [e]CON vs ICE-SS. *Large and [#]moderate effect size, ICE vs CON; [$]large and [¥]moderate effect size, ICE-SS vs CON-SS; [&]large and [§]moderate effect size, ICE-SS vs CON.

**Table 2. Whole body sweat loss during immersion and recovery.**

|  | ICE | CON | ICE-SS | CON-SS |
|---|---|---|---|---|
| Whole body sweat loss during immersion (L) | 0.92 ± 0.56 | 0.90 ± 0.59 | 0.96 ± 0.42 | 1.04 ± 0.52 |
| Whole body sweat loss during recovery (L) | 0.65 ± 0.26 | 0.69 ± 0.19 | 0.61 ± 0.30 | 0.62 ± 0.23 |

ICE, ice slurry, no sweat suit; CON, control drink, no sweat suit; ICE-SS; ice slurry, sweat suit; CON-SS, control drink, sweat suit.

## Discussion

The aim of this study was to assess the cooling effectiveness of ice slurry ingestion in a state of compromised heat loss after passive heat exposure. This is the first study to explore the recovery from hyperthermia during heat loss restriction using ice slurry ingestion. Our research found that ice slurry ingestion resulted in a significant reduction in heat storage and $T_{rec}$ when compared to ingesting a thermoneutral control drink, observed in both SS and no-SS trials. Despite overall comparable reductions in heat storage for SS and no-SS trials, we identified a delayed reduction in heat storage in SS trials. Additionally, a reduction in $T_{sk}$ was only observed in no-SS trials following ice slurry ingestion. These data suggest that ice slurry ingestion can enhance recovery from hyperthermia and furthermore may be an effective cooling strategy for those experiencing hyperthermia due to compromised heat loss.

The greater $T_{sk}$ during SS trials suggests the creation of a hot micro-environment that limited heat loss, which contributed to the greater heat storage in the SS trials compared with the no-SS trials. Analysis of heat storage is of particular interest, as it is suggested to be more reflective of overall tissue temperature changes within the body compared with $T_{rec}$ alone [28], hence representing a holistic measure of thermal strain. The overall reduction in heat storage for both SS and no-SS indicates that ice slurry ingestion in the recovery from hyperthermia can reduce thermal strain regardless of the impact of compromised heat loss. This is in line with the ice-slurry induced reduction in $T_{core}$ during exercise whilst wearing firefighter PPE [21] or raincoats [20], inducing a moderate rise in core temperature (~0.8–1.5˚C). Interestingly, whilst Alhadad et al. (2021) found an ice-slurry induced reduction in gastrointestinal temperature during low intensity exercise, they did not observe this during moderate intensity exercise, inducing a higher rise in core temperature (~2˚C). The authors suggest this to be likely due to the increased metabolic heat production and concluded ice slurry ingestion not to be effective in such conditions [20]. Whilst the present study investigated a similar 2˚C rise in core temperature, the temperature rise investigated did not result from metabolic heat production but passive exposure to heat, which may explain why ice slurry proved to be effective in recovery from hyperthermia in the present study, but not during exercise investigated previously [20].

The timing of the greatest ice slurry induced reduction of heat storage differed between SS and no-SS trials, the biggest difference found earlier for no-SS trials. It is likely that the rise in heat storage during the early phases of recovery contributed this response. In contrast to ICE, during ICE-SS, heat storage did hence not return to baseline levels at 60 min. Prolonged ingestion time and/or larger quantities of slurry in conditions of considerable thermal strain may therefore be required to more effectively reduce heat storage [29]. This would be of particular interest for occupations where heat loss is limited by clothing microenvironments such as military personnel, firefighters and motorsport athletes, and in populations with impaired sweating ability such as those with spinal cord injury [2, 30].

The ice slurry ingestion induced reduction in $T_{rec}$ aligns with previous research investigating ice slurry ingestion during passive recovery [10, 16, 29]. However, as the literature is ambiguous, the present results contrasts others. For example, Iwata et al. exposed participants

to a hot (~38˚C, ~40% RH, in which sweating is likely effective) and found no differences in $T_{rec}$ and a trend for lower sweat rate in response to ice slurry ingestion [17]. This finding led to the suggestion that ice slurry ingestion may reduce sweating—and thus the evaporative heat loss potential from the skin—by an amount that is at least equivalent to the additional internal heat loss from ice slurry ingestion [31]. The findings of the present study do not support this theory for three reasons. First, when participants were exposed to dry, ambient conditions in which evaporative heat loss was likely effective (no-SS trials), ice slurry ingestion in fact *did* reduce $T_{rec}$. Second, if in line with that theory, our results should show an even more pronounced ice-slurry induced reduction in $T_{rec}$ when heat loss was compromised in SS trials. However, this was not the case–whilst the moderate and large effect sizes imply that ice slurry ingestion also has the potential to reduce $T_{rec}$ in SS trials, the ice-slurry induced difference in $T_{rec}$ was not significant in SS trials but was significant in no-SS trials, likely explained by the evaporative heat loss from the skin in the no-SS condition. Third, no changes in sweat loss were found between conditions, suggesting that ice slurry ingestion *per se* did not impede evaporative cooling potential at the skin. Here we must point out that our findings are in contrast to previous studies that have reported lower sweat rates following ice slurry ingestion [15, 18], attributed to internal thermoreceptor activation, and hence potential for reduced evaporative heat loss [15]. In summary, the present results do not support the suggestion that ice slurry ingestion negatively affects heat dissipation at the skin [31]. In the present study, internal ice-slurry induced heat loss is likely the main mechanism to explain the moderate to large effect sizes for the reductions in heat storage and $T_{rec}$ for the SS trials during recovery in ambient conditions. Contrasting findings within the literature regarding cooling capacity of ice slurries can be explained with differences in ambient conditions (facilitating evaporation or not), the cooling potential of the slurry (amount given, temperature), as well as differences in the experimental setup (timing, measurement procedures).

Ice slurry ingestion induced reductions in mean $T_{sk}$ have been reported previously [10], even though $T_{sk}$ has also been shown to be unaffected by ice slurry ingestion [31, 32]. In the present study, moderate effect sizes document a reduction in mean $T_{sk}$ following ingestion of ice slurry in no-SS, which likely is the result of conduction, i.e., the blood surrounding the gastrointestinal area cooled by the ice slurry flowing to and cooling the skin. On the other hand, the lack of change in $T_{sk}$ in SS trials is likely explained by the hot and humid microenvironment created by the SS, limiting convection and sweat evaporation to contribute to heat balance, likely counteracting any slurry-induced cooling of the skin. Whilst most skin surface heat loss in hot environments occurs via evaporation, with any modification to sweating and the evaporation of sweat having the greatest influence on heat balance [31], the reduction in $T_{sk}$ therefore implies that conductive heat loss is a mechanism by which ice slurry ingestion exerts cooling.

The higher mean $T_{sk}$ in the SS trials further explain the observed increases in thermal sensation, a mechanistic link that is established for $T_{sk}$ [33]. Higher thermal sensation under conditions of compromised heat loss is also in line with previous research suggesting that individuals with impaired sweating ability (notably athletes with spinal cord injury) exhibit greater $T_{sk}$ and increased thermal sensation than those without impaired sweating ability [34]. Interestingly, no thermal sensation differences were observed between SS and no-SS trials when ice slurries were ingested, indicating that ice slurry ingestion mitigates the increased thermal sensation associated with compromised heat loss. In addition, the present results highlight the effectiveness of ice slurry ingestion in reducing thermal sensation in both SS and no-SS trials and hence benefit conditions of normal as well as compromised heat loss.

We must note that a potential limitation of the present approach to determine heat storage lies in its calculation based on a two-compartment model, using measures of $T_{rec}$ and $T_{sk}$

alone. Indeed, more accurate estimations of heat storage during exercise include a contribution of the muscle compartment [28]. However, the present study investigated the recovery following passive heating in a non-exercising state which does not increase muscle temperature due to muscular work. Nonetheless, the inclusion of the muscle compartment, or calorimetry-based approaches for heat storage estimations, may be considered for any follow-up studies.

## Conclusions

This study demonstrates that internal cooling using ice slurries can effectively reduce thermal strain following hyperthermia. Regardless of whether heat loss was compromised or not, ice slurry ingestion led to a decrease in heat storage. However, a delayed pattern for reducing markers of thermal strain was apparent in a state of compromised heat loss, which may be related to ice slurry ingestion not lowering $T_{sk}$ in this state. The reduced heat storage was accompanied by a reduced $T_{rec}$, highlighted by moderate to large effect sizes in both SS and no-SS trials. Our findings suggest ice slurry ingestion can enhance recovery from hyperthermia, and furthermore may be an effective cooling strategy for individuals wearing clothing that inhibits convection and evaporation and populations with compromised sweating ability.

## Supporting information

**S1 Data. Rectal and skin temperature, heat storage, thermal sensation and all resting measures used in this study.**
(XLSX)

## Acknowledgments

The authors would like to thank Sophie Goves, Chris Green, Boyu Duan, Jess Flint, Joe Craggs and Joseph Riley for assistance during data collection.

## Author Contributions

**Conceptualization:** Thomas J. O'Brien, Victoria L. Goosey-Tolfrey, Christof A. Leicht.

**Formal analysis:** Thomas J. O'Brien, Christof A. Leicht.

**Investigation:** Thomas J. O'Brien.

**Methodology:** Thomas J. O'Brien, Christof A. Leicht.

**Project administration:** Thomas J. O'Brien.

**Resources:** Victoria L. Goosey-Tolfrey.

**Supervision:** Victoria L. Goosey-Tolfrey, Christof A. Leicht.

**Visualization:** Thomas J. O'Brien.

**Writing – original draft:** Thomas J. O'Brien.

**Writing – review & editing:** Thomas J. O'Brien, Victoria L. Goosey-Tolfrey, Christof A. Leicht.

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
