## [Decision Letter · Decision Letter 0]

22 Dec 2023

PONE-D-23-22378Compromised heat loss leads to a delayed ice slurry induced reduction in heat storagePLOS ONE

Dear Dr. O'Brien,

Thank you for submitting your manuscript to PLOS ONE. After careful consideration, we feel that it has merit but does not fully meet PLOS ONE’s publication criteria as it currently stands. Therefore, we invite you to submit a revised version of the manuscript that addresses the points raised during the review process.

We look forward to receiving your revised manuscript.

Kind regards,

Alan Ruddock

Academic Editor

PLOS ONE

Journal Requirements:

"No external financial support has been received for the preparation of this work"

Additional Editor Comments:

At the end of your introduction please consider the use of the term and associated sentence ‘we hypothesise…’; strictly this is not the correct form of a hypothesis. If you would like to keep this term, please state the null hypothesis(es) that you tested. What you have presented here is an assumption which is the origin of a hypothesis and is probably a better term to use. For example, ‘We assumed that…’.  Please see reviewer 1's comments regarding the null hypothesis. 

Please restructure the start of your discussion. Consider: 

Begin by re-stating the aims of your studyHighlight the originality of your workState the significance of your findingsDiscuss the robustness of your research methodsSummarise your main findings

In your conclusion please make reference to the potential impact of your work, rather than suggesting your work might be of interest to a part of the scientific community. 

Methods - replace 'height' with 'stature'

Check spacing between units and associated metric e.g. 6.8g·kg-1 should have a space. Temperature should not have a space as you quite rightly format so (but please check). There are several instances of this incorrect formatting within the abstract. 

There are quite a lot of non-standard abbreviations. I would suggest you limit these to aid interpretation. 

Physiological measurements - change to physiological assessments

Inconsistent use of p = and p <; please clarify why you have reported p values in this way. I think you use < when you are referring to two or more outcomes but please indicate. 

Reviewer 2 notes that figures are pixelated - this is is to lower the file size not due to your formatting. Please disregard this comment. 

Overall this is a well written manuscript. The changes suggested to the structure of the paper will highlight the strengths of your work. Well done. 

Reviewers' comments:

Reviewer's Responses to Questions

**Comments to the Author**

1. Is the manuscript technically sound, and do the data support the conclusions?

Reviewer #1: Yes

Reviewer #2: Yes

2. Has the statistical analysis been performed appropriately and rigorously? 

Reviewer #1: Yes

Reviewer #2: I Don't Know

3. Have the authors made all data underlying the findings in their manuscript fully available?

Reviewer #1: Yes

Reviewer #2: Yes

4. Is the manuscript presented in an intelligible fashion and written in standard English?

Reviewer #1: Yes

Reviewer #2: Yes

5. Review Comments to the Author

Reviewer #1: Abstract

L32: How was this measured? Needs to be presented in the abstract

L41-43: How are the implications of these findings going to impact a particular population type. A short summary of population impact will improve this section

Introduction

L51: To balance the argument that heat exposure is not always a negative, include information highlighting the benefits of heat stress to exercise performance

L66-69: Whilst the literature differs the reader is not sure why this is the case. Further, information to highlight differences in study design (e.g. dosage) could be included here

L85: What are the practical situations where heat loss could be compromised? Provide examples here in this section

L89-91: Ensure hypothesis is testable. If null hypothesis significance testing has been performed, then it should state significantly reduce

Methods

L100: Was a power calculation performed?

L116-119: What was air temperature whilst the participants were in the bath? Was the temperature continuously measured?

L164: Can the authors rationalize as to why a post-hoc correction (e.g. bonferroni correction) was not used? As SPSS was used for the statistical analysis the authors could have used paste function to alter the code for the ANOVA and thus used a post-hoc test?

Results

Did the authors consider presenting the 95%CI alongside the p values to show these changes?

L178: Was an ANOVA performed on urine osmolality? Could this be included in table 1 if not?

Discussion

The discussion highlights the key areas of the study and is well written. However, the hypothesis must be accepted or rejected.

Reviewer #2: Aim: To assess whether ice slurry is effective at reducing thermal strain in temperate conditions when convective and evaporative heat loss is limited/compromised

Seems to be ICE vs CON and ICE-SS vs CON-SS. I’d be interested in ICE-SS vs CON

Heat storage? Or Core temp? Referring to starting point before immersion so countering heat storage from immersion (exercise in practice) – This is confusing throughout. “the greatest differences in ice slurry induced heat storage” implies heat being stored (increase) not dissipated.

Corresponding author say Thomas O’Brien at the top but Christof Leicht in the manuscript

Introduction

Looking at it from a clothing restriction/SCI point of view rather than slurry causing possible reduction in evaporative mechanism, is this something to consider? This is used as the rationale for using ice slurry in a population with compromised heat loss as evaporative potential is non-existent so cannot decrease from internal thermo-receptors.

You briefly mention this could be useful post-exercise due to potential practical limitations of implementing during sport. You have argued the benefit of ice slurry when heat loss mechanisms are compromised, what is the need for effective passive post-match recovery in temperate conditions?

As your study was recovery in temperate conditions, would this be applied to competing in heat then recovering in air conditioned room or exercising in PPE or with SCI in temperate environments and seeing high rates of heat storage due to compromised heat loss mechanisms?

Methods

Experimental design: I found the ordering of this paragraph a little confusing and wanted to see the detail on the target core temp increase before the sentence “followed by a 60-min recovery phase”. Likely personal preference here – the detail is there

Line 120: How did you achieve ice slurry consistency with no flavouring/additives. In my experience this would just be shards of ice with water. Please can you add the detail as to how you made your slurries.

Line 122: ingest within 15 minutes at 5 and 20 min – “at” should be “between”

Line 133: What was the threshold for determining adequate hydration?

How were the ibuttons secured? This could be saturated and affect assessment of sweat rate if nude body mass is instrumented.

Why did you define trend as p = 0.05-0.10?

Stats – why condition x drink x time and not just condition x time? (condition (4) x time (7))

Results

The picture quality of figure 1 is poor

Table 2 – I think pre-post immersion is confusing. I would align this with during recovery to be during immersion

Discussion

Line 258: 40% RH a humid environment not dry. Previous research has classed 40˚C 40% RH as hot humid. Sweating would still be effective but not the same as 0-20% RH

Line 259: Did Iwata et al. observe a reduced sweat rate whilst not seeing a reduction in core temp from ice slurry ingestion?

Line 265 and 266: I think this second point should be that the ICE and ICE-SS heat loss should be the same i.e. reduced evaporative potential due to proposed reduction in sweat rate. There is more heat loss in ICE compared to ICE-SS suggesting heat is dissipated evaporatively alongside ice slurry ingestion. They key difference would be rate of heat dissipation with 2 mechanisms rather than 1.

I don’t think your stats can show this interaction as you split your conditions in to drink and SS. You have presented a difference between conditions (ICE-SS and ICE) in figure 1A, but not the interaction of conditions over time Maybe this wasn’t presented because it wasn’t significant but, I think worth a mention as you talk about it in the discussion. I am not clear on how your stats allowed this assessment though

Line 276: The main mechanism doesn’t seem accurate. Internal heat loss was a mechanism but no-SS saw increased heat dissipation compared to SS (from looking at the graph) which was maybe due to evaporative heat loss.

Worth noting this is for recovery in a temperate environment but these levels of heat storage are observed when heat loss is compromised

6. PLOS authors have the option to publish the peer review history of their article (what does this mean?). If published, this will include your full peer review and any attached files.

Reviewer #1: **Yes: **Jeffrey Aldous

Reviewer #2: **Yes: **Matthew Debney

---

## [Author Response · Author response to Decision Letter 0]

20 Feb 2024

Journal Requirements:

"No external financial support has been received for the preparation of this work"

I have included at the bottom of a revised cover letter the following statement:

“This research was funded by the Peter Harrison Centre for Disability Sport (PHC), a research centre within the School of Sport, Exercise and Health Sciences at Loughborough University. The funders had no role in the study design, data collection and analysis, decision to publish, or preparation of the manuscript. The lead other, Thomas O’Brien was receiving a salary from the PHC at the time of submission.”

Additional Editor Comments:

At the end of your introduction please consider the use of the term and associated sentence ‘we hypothesise…’; strictly this is not the correct form of a hypothesis. If you would like to keep this term, please state the null hypothesis(es) that you tested. What you have presented here is an assumption which is the origin of a hypothesis and is probably a better term to use. For example, ‘We assumed that…’. Please see reviewer 1's comments regarding the null hypothesis. 

Thank you. We have altered this sentence in line with your feedback. 

Please restructure the start of your discussion. Consider: 

• Begin by re-stating the aims of your study

• Highlight the originality of your work

• State the significance of your findings

• Discuss the robustness of your research methods

• Summarise your main findings

The first paragraph of our discussion has now been re-written to reflect the considerations presented above. We feel that elements relating to the robustness of our study are presented throughout. For example, controlling participant body temperature change, ensuring constant water temperature r (39.4-39.6℃), ensuring comparable reduction in evaporative potential across participants by wearing a sweat suit, relativising drink quantity to body mass, and supplementing significant findings with effect sizes. We feel that we cannot summarise these (and more) factors relating to robustness in a meaningful and succinct form within this first ‘summary’ paragraph and feel that if we tried, it would tip the balance from summarising the main findings into a defence of the experimental approach.

In your conclusion please make reference to the potential impact of your work, rather than suggesting your work might be of interest to a part of the scientific community. 

Thank you for your comment. We have altered our concluding sentence to highlight the practical applications.

Methods - replace 'height' with 'stature'

This has been corrected on line 103.

Check spacing between units and associated metric e.g. 6.8g·kg-1 should have a space. Temperature should not have a space as you quite rightly format so (but please check). There are several instances of this incorrect formatting within the abstract. 

This has been corrected throughout.

There are quite a lot of non-standard abbreviations. I would suggest you limit these to aid interpretation. 

Thank you for your comment. Whilst we recognise we have used some non-standard abbreviations, they are key to keeping our manuscript concise. For example. Writing “Ice slurry, no sweat suit”, “control drink, no sweat suit”, “Ice slurry, sweat suit”, and “control drink, sweat suit” in full at every mention would make sentences hard to read. All other abbreviations are standard within the literature.

Physiological measurements - change to physiological assessments

This has been changed on line 139.

Inconsistent use of p = and p <; please clarify why you have reported p values in this way. I think you use < when you are referring to two or more outcomes but please indicate. 

This has been included in our statistical analysis section on line 159-160.

Reviewer 2 notes that figures are pixelated - this is is to lower the file size not due to your formatting. Please disregard this comment. 

Thank you.

Overall this is a well written manuscript. The changes suggested to the structure of the paper will highlight the strengths of your work. Well done. 

Thank you for your kind words. 

Reviewers' comments:

Reviewer's Responses to Questions

Comments to the Author

Reviewer #1: Abstract

L32: How was this measured? Needs to be presented in the abstract

The two-compartment thermometry model of heat storage was used which has been added to the abstract. Further, specific details of this are found within the manuscript on line 150.

L41-43: How are the implications of these findings going to impact a particular population type. A short summary of population impact will improve this section

This has been added to the end of the first conclusion sentence of the abstract.

Introduction

L51: To balance the argument that heat exposure is not always a negative, include information highlighting the benefits of heat stress to exercise performance

A sentence highlighting this has been added to the start of the paragraph.

L66-69: Whilst the literature differs the reader is not sure why this is the case. Further, information to highlight differences in study design (e.g. dosage) could be included here

This has been added to the end of the paragraph. 

L85: What are the practical situations where heat loss could be compromised? Provide examples here in this section

This has already been detailed in the opening paragraph of the introduction and therefore would be repetitive. These situations include individuals wearing clothing such as PPE, or those with disrupted neural pathways such as individuals with spinal cord injury. 

L89-91: Ensure hypothesis is testable. If null hypothesis significance testing has been performed, then it should state significantly reduce

Thank you for your comment. This has now been altered based on your feedback and the editors comment.

Methods

L100: Was a power calculation performed?

No we didn’t perform a power calculation. Given our main findings showing the effectiveness of ice slurry in both normal heat loss and compromised heat loss states, the study was powered adequately.

L116-119: What was air temperature whilst the participants were in the bath? Was the temperature continuously measured?

Air temperature was measured at the start of each trial only, which was not significantly different across trials. It was a potential limitation that we didn’t measure air temperature in proximity of the participants’ head in the bath throughout. However, as we had a cut off for rectal temperature, participants achieved this independent of time.

L164: Can the authors rationalize as to why a post-hoc correction (e.g. bonferroni correction) was not used? As SPSS was used for the statistical analysis the authors could have used paste function to alter the code for the ANOVA and thus used a post-hoc test?

We used a 2x2x7 repeated measures ANOVA and reported main and interaction effects. This was then followed up by distinct (separate) ANOVAs for each timepoint, therefore the suggested approach would not apply (it would apply to the overall ANOVA, however, as we only report main and interaction effects for these without conducting post-hoc tests no such results are presented – given the complex nature of the ANOVA (2x2x7) such post-hoc tests would not allow meaningful discussion). The only way to apply a Bonferroni post-hoc correction in our chosen approach would be to divide the critical p value by the number of tests conducted (7 time points). We chose not to apply Bonferroni corrections for these comparison as this would have increased the likelihood of type II errors (Perneger, 1998). To respond to critics of this approach, we also supplemented all comparison with effect sizes in addition to P values to provide a more rounded picture. 

Results

Did the authors consider presenting the 95%CI alongside the p values to show these changes?

We indeed considered presenting CI within our results, however due to the multiple comparisons we feel that this would go at the expense of clarity by overloading this section with numbers. We feel that presenting p values, supported by effect sizes, offers adequate information regarding the presented comparisons, and that presenting CI would add very little extra information but negatively impact readability. 

L178: Was an ANOVA performed on urine osmolality? Could this be included in table 1 if not?

This has been added into the statistical analysis section and table 1 of the results.

Discussion

The discussion highlights the key areas of the study and is well written. However, the hypothesis must be accepted or rejected.

Thank you for your comment. We have adjusted our original hypothesis statement within the introduction in line with editors comments, and therefore this is no longer needed. 

Reviewer #2: Aim: To assess whether ice slurry is effective at reducing thermal strain in temperate conditions when convective and evaporative heat loss is limited/compromised

Seems to be ICE vs CON and ICE-SS vs CON-SS. I’d be interested in ICE-SS vs CON

We chose to use a three-way ANOVA and therefore this comparison is not advised. Whilst we understand interpretations from a three-way ANOVA can be difficult without a three-way interaction, with a two-way interaction one of the factors remain combined (E.G. slurry and sweat suit), therefore it would be incorrect or inadvisable to perform a time point comparison between all four conditions. 

Heat storage? Or Core temp? Referring to starting point before immersion so countering heat storage from immersion (exercise in practice) – This is confusing throughout. “the greatest differences in ice slurry induced heat storage” implies heat being stored (increase) not dissipated.

This has been edited on line 276 to clarify we are referring to the ice slurry induced reduction in heat storage. We have checked the manuscript throughout and now consistently apply this wording.

Corresponding author say Thomas O’Brien at the top but Christof Leicht in the manuscript

This is made clear on the title page of the manuscript and has been changed online. 

Introduction

Looking at it from a clothing restriction/SCI point of view rather than slurry causing possible reduction in evaporative mechanism, is this something to consider? This is used as the rationale for using ice slurry in a population with compromised heat loss as evaporative potential is non-existent so cannot decrease from internal thermo-receptors.

Thank you for your comment. We have indeed referred to the clothing restriction/SCI point of view on lines 73-77 in our introduction, stating that ice slurry may be of particular interest when sweating does not contribute to heat loss (such as clothing restriction and SCI).

You briefly mention this could be useful post-exercise due to potential practical limitations of implementing during sport. You have argued the benefit of ice slurry when heat loss mechanisms are compromised, what is the need for effective passive post-match recovery in temperate conditions?

The need for effective passive post-match recovery in temperate conditions include when competing in the heat with multiple matches or events per day. Further, when seeing continued high levels of heat storage due to compromised heat loss such as in SCI or when wearing PPE which could potentially lead to reaching critical core temperatures. An example of this in persons with SCI competing at the Paralympics, where high levels of heat storage are witnessed during game play and beyond with the potential expectation to play multiple games per day. 

As your study was recovery in temperate conditions, would this be applied to competing in heat then recovering in air conditioned room or exercising in PPE or with SCI in temperate environments and seeing high rates of heat storage due to compromised heat loss mechanisms?

Yes, our results may also inform strategies around individuals competing in the heat and recovering in air-conditioned environments when hyperthermia is experienced. However, it is important to note that the method of inducing hyperthermia in our study was hot water immersion, not exercise in the heat so caution should be taken. 

Methods

Experimental design: I found the ordering of this paragraph a little confusing and wanted to see the detail on the target core temp increase before the sentence “followed by a 60-min recovery phase”. Likely personal preference here – the detail is there

Thank you for your suggestion. We have changed the order of this paragraph. 

Line 120: How did you achieve ice slurry consistency with no flavouring/additives. In my experience this would just be shards of ice with water. Please can you add the detail as to how you made your slurries.

Small ice cubes were shaved using a commercially available instrument before being blended with cold water to achieve an ice slurry consistency. This detail has been added to the methods on line 123.

Line 122: ingest within 15 minutes at 5 and 20 min – “at” should be “between”

We asked participants to ingest ice slurry at 5 minutes and at 20 minutes of recovery and gave them 15 minutes to do so each time, so feel this change is not warranted. 

Line 133: What was the threshold for determining adequate hydration?

The threshold for determining adequate hydration prior to the start of the test was <700 mOsm·kg-1 as highlighted by the ACSM position stand (Sawka et al., 2007).

How were the ibuttons secured? This could be saturated and affect assessment of sweat rate if nude body mass is instrumented.

Ibuttons were secured using 3M Transpore surgical tape which is water resistant. 

Why did you define trend as p = 0.05-0.10?

In our results we refer to “approaching significance”, and therefore we provide transparency in our interpretation of “approaching significance”. 

Stats – why condition x drink x time and not just condition x time? (condition (4) x time (7))

We chose to use a three-way ANOVA as when performing a two-way ANOVA one of the factors remain combined (E.G. slurry and sweat suit), and therefore it would be incorrect and inadvisable to perform a time point comparison between all four conditions. 

Results

The picture quality of figure 1 is poor

Please see editor comment above.

Table 2 – I think pre-post immersion is confusing. I would align this with during recovery to be during immersion

Thank you. This has been changed.

Discussion

Line 258: 40% RH a humid environment not dry. Previous research has classed 40?C 40% RH as hot humid. Sweating would still be effective but not the same as 0-20% RH

Thank you for your observation, this description of ‘dry’ environment has been taken out. 

Line 259: Did Iwata et al. observe a reduced sweat rate whilst not seeing a reduction in core temp from ice slurry ingestion?

Whole body sweat rate was not significantly different between trials but trended lower in the ICE trials (ICE = 454.3 ± 172.3 g, CON = 539.7 ± 157.2 

---

## [Decision Letter · Decision Letter 1]

23 May 2024

Compromised heat loss leads to a delayed ice slurry induced reduction in heat storage

PONE-D-23-22378R1

Dear Dr. Leicht,

We’re pleased to inform you that your manuscript has been judged scientifically suitable for publication and will be formally accepted for publication once it meets all outstanding technical requirements.

Kind regards,

Alan Ruddock

Academic Editor

PLOS ONE

Reviewers' comments:

Reviewer's Responses to Questions

**Comments to the Author**

1. If the authors have adequately addressed your comments raised in a previous round of review and you feel that this manuscript is now acceptable for publication, you may indicate that here to bypass the “Comments to the Author” section, enter your conflict of interest statement in the “Confidential to Editor” section, and submit your "Accept" recommendation.

Reviewer #1: All comments have been addressed

2. Is the manuscript technically sound, and do the data support the conclusions?

Reviewer #1: Yes

3. Has the statistical analysis been performed appropriately and rigorously? 

Reviewer #1: Yes

4. Have the authors made all data underlying the findings in their manuscript fully available?

Reviewer #1: Yes

5. Is the manuscript presented in an intelligible fashion and written in standard English?

Reviewer #1: Yes

6. Review Comments to the Author

Reviewer #1: (No Response)

7. PLOS authors have the option to publish the peer review history of their article (what does this mean?). If published, this will include your full peer review and any attached files.

Reviewer #1: **Yes: **Dr Jeffrey Aldous

---

## [Editor Report · Acceptance letter]

1 Aug 2024

PONE-D-23-22378R1 

PLOS ONE

Dear Dr. O'Brien, 

I'm pleased to inform you that your manuscript has been deemed suitable for publication in PLOS ONE. Congratulations! Your manuscript is now being handed over to our production team.

Kind regards, 

on behalf of

Dr. Alan Ruddock 

Academic Editor

PLOS ONE